# Valve-in-Valve Transcatheter Aortic Valve Replacement: From Pre-Procedural Planning to Procedural Scenarios and Possible Complications

**DOI:** 10.3390/jcm13020341

**Published:** 2024-01-07

**Authors:** Francesca Maria Di Muro, Chiara Cirillo, Luca Esposito, Angelo Silverio, Germano Junior Ferruzzi, Debora D’Elia, Ciro Formisano, Stefano Romei, Maria Giovanna Vassallo, Marco Di Maio, Tiziana Attisano, Francesco Meucci, Carmine Vecchione, Michele Bellino, Gennaro Galasso

**Affiliations:** 1Structural Interventional Cardiology, Department of Clinical and Experimental Medicine, Clinica Medica, Careggi University Hospital, 50134 Florence, Italy; francescamaria.dimuro@unifi.it (F.M.D.M.); meuccif@aou-careggi.toscana.it (F.M.); 2Oxford Heart Centre, Oxford University Trust, Oxford OX3 9DU, UK; 3Department of Medicine, Surgery and Dentistry, University of Salerno, Baronissi, Salvador Allende Street 43, 84081 Salerno, Italy; lesposito@unisa.it (L.E.); c.formisano5@studenti.unisa.it (C.F.); sromei@unisa.it (S.R.); ggalasso@unisa.it (G.G.); 4Department of Advanced Biomedical Sciences, University Federico II, 80138 Naples, Italy; 5Division of Cardiology, Cardiovascular and Thoracic Department, San Giovanni di Dio e Ruggi d’ Aragona University Hospital, 84131 Salerno, Italy; tiziana.attisano@sangiovannieruggi.it

**Keywords:** bioprosthetic heart valves, structural valve deterioration, TAVR, pre-procedural planning

## Abstract

Over the last decades, bioprosthetic heart valves (BHV) have been increasingly implanted instead of mechanical valves in patients undergoing surgical aortic valve replacement (SAVR). Structural valve deterioration (SVD) is a common issue at follow-up and can justify the need for a reintervention. In the evolving landscape of interventional cardiology, valve-in-valve transcatheter aortic valve replacement (ViV TAVR) has emerged as a remarkable innovation to address the complex challenges of patients previously treated with SAVR and has rapidly gained prominence as a feasible technique especially in patients at high surgical risk. On the other hand, the expanding indications for TAVR in progressively younger patients with severe aortic stenosis pose the crucial question on the long-term durability of transcatheter heart valves (THVs), as patients might outlive the bioprosthetic valve. In this review, we provide an overview on the role of ViV TAVR for failed surgical and transcatheter BHVs, with a specific focus on current clinical evidence, pre-procedural planning, procedural techniques, and possible complications. The combination of integrated Heart Team discussion with interventional growth curve makes it possible to achieve best ViV TAVR results and avoid complications or put oneself ahead of time from them.

## 1. Introduction

Over the last decades, bioprosthetic heart valves (BHV) have been increasingly implanted instead of mechanical valves in patients undergoing surgical aortic valve replacement (SAVR).

Deciding between a mechanical and biologic aortic prosthesis is a daily discussion between the patient and surgeon. Data in the literature are controversial, with no unanimous watershed on indication for type of prothesis to implant in candidates for surgery. Indeed, American Heart Association (AHA) Guidelines [1] recommend mechanical aortic prosthesis in the absence of contraindications to anticoagulation in patients younger than 50 years of age (Class of Recommendation COE 2a, Level of evidence LOE B-R). This threshold raises to 60 years in the European Guidelines (COR IIa LOE B) [2].

Although the main obstacle to mechanical prostheses is the difficult management of a life-long warfarin therapy, often contraindicated in some patients [3], BHVs tend to fail early and often requires reintervention [4]. Considering the progressive decrease in the age of patients treated with BHVs, valve durability will be a major clinical issue in cardiovascular medicine in the upcoming years. Indeed, compared to primary SAVR, redo-SAVR is associated with higher morbidity and mortality [5].

On the other side, TAVR is established for the treatment of severe symptomatic aortic stenosis (AS) in patients at high surgical risk and recently showed favorable outcomes in patients at lower surgical risk [6,7]. In the evolving landscape of interventional cardiology, (ViV) TAVR has emerged as a remarkable innovation to address the complex challenges of patients previously treated with SAVR and has rapidly gained prominence as a feasible technique in the management of failed surgical BHVs. ViV TAVR is considered to be a safe and effective therapeutic option in patients at high surgical risk [1,2]. However, the growing use of TAVR in younger patients with severe AS raises concerns on the long-term durability of THVs, as patients might outlive the bioprosthetic valve [8,9].

In this review, we provide an overview on the role of ViV TAVR for failed surgical and transcatheter BHVs, with a specific focus on current clinical evidence, pre-procedural planning, procedural techniques, and possible complications.

## 2. ViV TAVR vs. Redo-SAVR: Evidence, Indications and Patients’ Selection

To date, no randomized clinical trial (RCT) has compared ViV TAVR and redo-SAVR in patients with failed surgical BHVs. Several observational studies showed the feasibility of ViV TAVR with both balloon-expandable (BE) and self-expandable (SE) THVs in patients who underwent previous SAVR with BHVs [10,11,12,13]. In an observational study from the French administrative database, ViV TAVR was associated with significantly lower rates of the composite of all-cause mortality, all-cause stroke, myocardial infarction (MI), and major or life-threatening bleeding at 30 days compared to redo-SAVR. However, at a median follow-up of 516 days, there were no significant differences between the two groups in terms of the composite outcome of cardiovascular death, all-cause stroke, MI or rehospitalization for heart failure [14]. In a propensity score-matched analysis on patients undergoing intervention for failed BHVs with either ViV TAVR or redo-SAVR, Tam et al. found that ViV TAVR was associated with lower rates of 30-day mortality, lower length of in-hospital stay, and increased 5-year survival (76.8% vs. 66.8%) compared to redo-SAVR [15]. A recent metanalysis of 12 observational studies and 8430 patients showed that ViV TAVR was associated with similar risks of all-cause mortality, cardiovascular mortality, MI, permanent pacemaker implantation and moderate to severe paravalvular leaks (PVL) compared to redo-SAVR at a median follow-up of 1.74 years; however, significantly lower rates of major bleeding, stroke, procedural mortality, and 30-day mortality were reported in patients treated with ViV TAVR [16].

Overall, even in absence of RCT, these data indicate ViV TAVR as a safe and effective strategy for patients deemed at high or prohibitive surgical risk.

Redo-SAVR is still considered in peculiar conditions:-patients at intermediate or low surgical risk;-young individuals with longer life expectancy (because no data are available on long-term durability of ViV TAVR);-patients with complex anatomical features for ViV TAVR, such as a high risk for coronary obstruction (without possibility of performing BASILICA) or with small anatomies.-in cases of non-structural BHV dysfunction, such as patient-prosthesis mismatch (PPM) or severe PVL (percutaneous approach might be reasonable in cases of PPM when a balloon valve fracture might be performed within a stented surgical valve or in cases of PVL suitable for a percutaneous closure).

## 3. Lifetime Management of Patients with AS: TAV Surgical Explant vs. Redo-TAVR for Failed THVs

Over the last years, TAVR has been increasingly adopted for the treatment of severe AS, with a progressive decrease in the mean age and surgical risk of patients [8]. Given the longer life expectancy of these patients, the incidence of THV failure is likely to increase in the next years.

The main treatment options for failed THVs are TAVR surgical explant and redo-TAVR within the previously implanted THV. The choice between the two treatment modalities should be tailored to the individual anatomical and clinical characteristics of the patient, including mechanism and timing of THV failure, aortic root anatomy, risk of coronary obstruction, type of failed THV and surgical risk profile.

In certain circumstances, such as endocarditis or severe PPM, redo-TAVR is not feasible and TAVR surgical explant is the only therapeutic option. TAVR explant is technically more complex and has a higher rate of perioperative complications compared to more common SAVR. During the procedure, the surgeon needs to remove the native aortic leaflets and the THV frame, with the possible risk of damage to the adjacent structures, such as aortic root, ascending aorta and mitral valve. The type of THV also influences technical complexity and the risk of complications, as THV with a longer frame extending into the ascending aorta may require more extensive surgical manipulations. The clinical outcome of patients treated with TAVR explant has been assessed in real-world studies. In an U.S. registry-based study on patients treated with TAVR between 2012 and 2017 using the Center for Medicare and Medicaid Services database, the 30-day and 1-year mortality rates of TAVR explant were 13.2% and 22.9%, respectively [17]. In a recent analysis from the Society of Thoracic Surgeons database on 123 patients treated with TAVR explant from 2011 to 2015, the operative mortality rate was 17.1%, with an increased observed-to-expected mortality [18]. The EXPLANT-TAVR registry assessed the clinical outcome of 269 patients who underwent TAVR explant between 2009 and 2020 and a high incidence of mortality at 30 days (13.1%) and 1 year (28.5%) was reported. Of note, the incidence of stroke at 30 days and 1 year was 8.6% and 18.7%, respectively. Taken together, these data show that TAVR explant is a complex surgical procedure, with a high risk of mortality at short and midterm [19].

Redo-TAVR is a reasonable therapeutic strategy in patients with THV failure, especially in those with SVD. It is an area of ongoing research, but, nowadays, the data on long-term clinical outcomes are still scarce. There are two main observational studies assessing the safety and efficacy of redo-TAVR in the real world. In an analysis from the Medicare database on 617 patients undergoing redo-TAVR in the U.S. between 2012 and 2017 (0.46% of all TAVR procedures), the mortality rates at 30 days and 1 year were 6.0% and 22.0%, respectively. Compared to a matched group of patients undergoing TAVR explant, redo-TAVR was associated with a lower 30-day mortality (6.2% vs. 12.3%), although 1-year mortality was similar in the two groups (21.0% vs. 20.8%). The incidence of major adverse cardiovascular events was lower in patients treated with redo-TAVR (risk ratio: 2.92) [20]. In the Redo-TAVR international registry, 212 consecutive patients treated with redo-TAVR were included. Patients were divided into two groups based on the timing of THV failure, within or after 1 year from the index procedure. Interestingly, survival rates at 30 days were acceptable, with no significant differences between the two groups (94.6% and 98.5% at 30 days; 83.6% and 88.3% at 1 year) [21]. However, these studies have some limitations. SVD was not the only mechanism of THV failure of patients enrolled. Moreover, the median interval from the index procedure to redo-TAVR was relatively short in both studies, about 5 months in the Medicare database and 33 months in the Redo-TAVR international registry [20,21].

As the incidence of THV failure is expected to rise in the next years, collecting long-term data on redo-TAVR and optimizing the planning of the first THV implantation will be crucial for refining the procedure and understanding its efficacy. This data can help to identify trends, complications, and areas for improvement. Continual monitoring, research and innovation in the field of transcatheter heart valve replacement will play a pivotal role in shaping the future management of THV failure.

## 4. Pre-Procedural Planning

### 4.1. TAVR in SAVR

The main limitations of ViV TAVR are related to a lack of space in the aortic root or to the mechanical complications potentially deriving from the deflection of surgical bioprosthetic valve leaflets. The former lead to high gradients and significant prosthesis-patient mismatch (PPM) and the latter to coronary obstruction or disruption [22]. Echocardiography and ECG-gated computer tomography (ECG-gated CT) allow us to assess surgical bioprosthetic valve dysfunction and the best planning for redo intervention.

According to the Academic Research Consortium (VARC)-3 criteria, the main aetiologies for failure of surgical bioprosthetic heart valves are structural valve deterioration (SVD), non-structural valve disfunction comprehensive of PVL and PPM, valve thrombosis and endocarditis [23].

Structural valve deterioration (SVD) is defined as intrinsic permanent changes to the prosthetic valve, including wear and tear, leaflet disruption, flail leaflet, leaflet fibrosis and/or calcification, and strut fracture, manifested as stenosis and/or regurgitation [23].

ViV TAVR is typically considered in patients with SVD rather than non-structural valve dysfunction, although patients with severe PPM with suitable anatomy may be considered for ViV TAVR with adjunctive procedures.

When planning a TAVR in SAVR it is essential to ascertain the model, size, and mechanism of the failing surgical valve.

Surgical aortic bioprostheses can be classified based on their material (bovine or porcine) or on the frame type (stented, stentless or sutureless) as shown in Table 1.

Bovine pericardial valves tend to develop stenosis, while porcine valves tend to develop regurgitation [24]. Each model comes with a manufacturer labelled size which, in most cases, does not represent the true internal diameter (ID) of the prosthesis [25]. Stented bioprostheses usually have internally mounted leaflets generating a smaller true inner diameter (ID) than the one reported on the label and this can lead to oversize the ViV-TAVR. To overcome this issue, newer stented bioprostheses have externally mounted leaflets aimed to improve the haemodynamic profile and increase the true ID, but they carry a higher risk of coronary obstruction when deploying the ViV-TAVR.

Conversely, stentless bioprostheses have less risk of higher gradients, but the lack of fluoroscopic markers increases the risk of malpositioning, leaks and coronary obstruction.

Once the type and size of the valve is ascertained, a pre-ViV ECG-gated CT is essential to define the position of the valve in respect to the aortic valve plane, the true inner diameter, details on the aortic root/coronary ostia and the burden of calcification. The upper spots of the stent can be used to plan the best fluoroscopic view.

Current software can virtually place a cylinder in the ideal ViV TAVR position, allowing a calculation of the THV to coronary ostia distance (VTC). Patients with a VTC distance ≤ 4 mm are at increased risk of coronary obstruction and a cutoff of ≤ 3 mm is considered high risk. Other risk factors for coronary obstruction are narrow sinotubular junction/sinuses of Valsalva, previous root repair, and a supraannular position of the SAV [26].

A valve-in-valve international data (VIVID) classification of aortic root anatomy in THV-in-SAV has been proposed, together with a decision-making algorithm to guide procedural planning and need for BASILICA (leaflet splitting pre deployment) [27].

An indispensable tool to assess the suitability of each patient for a ViV and aid planning is the ‘ViV Aortic’ app, developed by Dr Vinayak Bapat. (https://www.pcronline.com/PCR-Publications/PCR-mobile-apps/Valve-in-Valve-Aortic-app, accessed on 22 November 2023)

Figure 1 and Figure 2 illustrate key moments in pre-procedural planning for TAVR in SAVR. See also Appendix A.

### 4.2. TAVR in TAVR

Preprocedural planning of a redo-TAVR requires a detailed analysis of the failed THV characteristics and patients’ anatomy. Current data are encouraging, but more data are needed on the best approach to plan and perform redo-TAVR [28]. Redo-TAVR is indicated for SVD according to the Valve Academic Research Consortium-3 definitions cited above [23]. As per surgical valves, careful analysis of the index THV characteristics, mechanism of failure and a detailed assessment of the patient anatomy is needed, as summarized in Table 2 [29] Echocardiography provides information on the failing mechanism of the valve, aortic root anatomy, and associated cardiac lesions.

The specific index THV dimensions given by manufacturers in the technical sizing chart are helpful to plan redo TAVR, but only ECG-gated CT will give the actual dimensions specific to the patient.

THV can be classified in balloon-expandable valves (BEV), mechanically expandable valves (MEV), and self-expanding valves (SEV). BEV and MEV have low stent frames and their leaflets are in intra-annular position, whereas SEV may have low or high stent frames and their leaflets are intra-annular and supra-annular, respectively.

When the redo THV is deployed, the failed THV leaflets are ‘pinned open’ creating a ‘neoskirt’. The neoskirt height will depend on the length of the stent frame and the height of the implant. These technical aspects have implications for the risk of coronary obstruction and must be considered when choosing the valve to use [30].

High risk features for TAVR in TAVR, such as endocarditis/valve thrombosis, severe PPM or significant paravalvular leaks not amenable of percutaneous treatment, and presence of other cardiac lesions needing cardiac surgery can be assessed in transthoracic and transoesophageal echocardiography [30].

Detailed analysis of both the pre-Index THV ECG-gated CT and a repeat ECG-gated CT Pre-Redo procedure are essential to planning.

Reanalysis of the pre-index THV CT is helpful to assess morphology and dimensions of the annulus, number of cusps, STJ and position in relation to coronary ostia. Reviewing the presence of unfavorable characteristics such as high burden of calcification, eccentricity, bicuspid aortic valve is essential.

The Pre-Redo-TAVI CT Analysis is needed to assess the aortic root anatomy, failing mechanism of the implanted valve, the true ID, stent, and skirt diameters and the position of the valve in relation to coronary ostia (risk of obstruction/sinus sequestration/difficulties accessing ostia) [31].

Risk factors for coronary obstruction/disruption are related to the index TAVI characteristics (neoskirt height, high/low implant, alignment with commissures, hypoexpansion) and to the characteristic of the redo valve of choice.

Figure 3 and Figure 4 show some critical pre-procedural planning stages for TAVR in TAVR.

## 5. Procedural Scenarios and Possible Complications Management

Optimal procedural execution, following a step-by-step codified approach, is crucial for the success of ViV procedures.

Most of ViV TAVR can be performed under conscious sedation with local anesthesia at site of puncture. Femoral access is preferred in most of the anatomies with 5 mm accepted as the minimum lumen diameter for the smallest transcatheter aortic valves (THV). A lumen diameter lower than 5 mm is associated with increased risks and the risks raise steeply when the sheath-to-femoral-artery ratio (SFAR) is more than 1.05 [32].

Regardless of the artery size, one should carefully consider the extent and the distribution of arterial wall calcifications. When the arterial wall is elastic with minor calcifications, a SFAR of 1.2 is acceptable and does not carry major risks. On the other hand, when the arterial wall is highly calcific, especially with circumferential distribution of calcium, the SFAR cut-off becomes less permissive. In this case, intra-vascular lithotripsy (IVL) assisted transfemoral TAVR has shown excellent results even if with slightly higher incidence of periprocedural complications [33].

When percutaneous femoral access is not feasible at all, trans-subclavian (TS), trans aortic (TAO), trans-apical (TA), trans-caval (TC) and trans-carotid routes might be considered [34].

A secondary vascular access, typically radial or contralateral femoral artery, is also required for basal aortography or for cannulation when coronary artery protection is needed (potentially with a stand-by stent on a coronary protection wire).

In case of SHV, the latter is crossed with Terumo straight wire supported by an AL1 or a JR4 catheter, according to different anatomies, then changed with a Safari2™ Pre-Shaped TAVI Guidewires (Boston Scientific, Miami, FL, USA) which allows enough support to advance the new THV.

Sometimes advancing the THV in the SHV can be challenging, especially when the SHV is stenotic. Different techniques and maneuvers to facilitate advancing the THV have been described such as: (A) the “buddy wire technique”, placing two safari wires in the LV to give more support to the advancing device; (B) the “buddy balloon technique” in case of calcific or tortuous segment, if a crystal balloon is used as a buddy to provide taper to the distal end of the delivery system and facilitate crossing; (C) the “snare technique” also described as the chaperone technique [35], when a snare is inserted through the contralateral femoral artery capturing the safari wire to achieve better alignment of the delivery system within the SHV (Figure 5).


**Deployment**


Once crossed the SHV, it is crucial to identify landmarks for optimal deployment of the new prosthesis.

Every surgical valve has its own design and dimensions, as well as fluoroscopic appearance; the sewing ring of the stented bio-prosthesis, sutured to the native aortic annulus, provides the most reliable rigid anchor to hold the THV, so it is important to know the relationship between the fluoroscopic markers and sewing ring location. As previously mentioned, the ViV Aortic app by Dr Vinayak Bapat helps the interventionalist by showing the angiographic aspect of each SHV and proposing a target for implantation with video examples [36].

In general, when using stented valves such as the SAPIEN valve, the aim should be to place its lowest profile 10–20% below the sewing ring of the SHV, while for the CoreValve, the limit is at least 4–6 mm below the ring.

Conversely, stentless THVs lack the rigid support of the sewing rings so the original suture line between SHV and native aortic annulus is still used as an anchor for THV. A slight oversizing of the prosthesis is usually required to achieve secure sealing. None of the commercially available stentless valves are radioopaque, which makes the implantation more challenging. Hence, techniques such as the placement of multiple pigtail catheters at the base of the leaflets, multiple contrast injections, and placement of a wire in the left main coronary artery are useful. Controlled deployment and the use of retrievable devices have facilitated this procedure because it is possible to try several times before completely release the THV.

## 6. Main Concerns to Consider during VIV Procedures


**Patient prosthesis mismatch (PPM)**


PPM can be considered the “Achilles’ heel” of VIV procedures and it is more frequent with small SHV, such as Mitroflow 21 mm (Sorin Group), Mosaic 19 mm and 21 mm (Medtronic, Minneapolis, MN, USA), TrifectaTM 21 mm (St. Jude Medical, St. Paul, MN, USA).

These cases have to be widely discussed in Heart Team and when VIV procedure remains the only option, there are some useful tips and tricks to consider for minimizing the risk of severe PPM:-**Self-expanding THV with supra-annular design:** When feasible, are the preferred choice in small SHV. Rodés-Cabau et al. [37], in their randomized trial comparing balloon vs. self-expanding valve systems in failed small SHV, founded an association between supra-annular SEV and improved hemodynamics with lower PPM.-**Higher THV deployment** (even if with a mild higher risk of coronary obstruction or valve embolization): As suggested in the VIVID registry [27] and confirmed by Simonato et al. [38] with their in vitro analysis. Considering the two most used THVs, the optimal implantation depth is 0 to 5 mm for the Corevalve and 0–2 mm up to 3.5 mm (0%–10% device frame) for Sapien THV.-**Bioprosthetic valve fracture (BVF) and bioprosthetic valve remodeling (BVR) [39]:** By either fracturing or stretching the surgical valve ring, providing increased THV expansion, better sealing, and post implantation hemodynamics. BVF is performed using a non-compliant balloon such as True Dilatation or Atlas Gold (Bard, Murry Hill, NJ, USA); (II) a high-pressure stopcock and tubing; (III) an indeflator and; (IV) a 60-mL syringe with dilute contrast. After initiating rapid ventricular pacing, the non-compliant balloon is rapidly inflated with 60 mL dilute contrast until fracture occurs. The best confirmation that BVF has occurred is by (A) angiographic modification of balloon waist and THV geometry, (B) a pressure drop in the in deflator or (C) an audible click concomitantly with the fracture. The balloon size should be determined by the THV used, the true ID of the SHV and the desired increase in diameter after fracture, the anatomy of the aortic root and LVOT and the height of coronary arteries. Generally, the balloon size should be 1 mm higher than the SHV diameter.-The timing of BVF, either before or after TAVR, remains controversial. BVF after THV implantation leads to better hemodynamics but carries a risk of damaging new prosthesis; BVF before THV implantation ensures better sealing, but may cause embolization of SHV, acute valvular regurgitation and hemodynamic instability [40,41]. The general practice is to do BVF after THV if using BEV so the NC balloon simultaneously fully expand the THV and fracture the surgical prosthesis while SEVs may not have enough force to fully expand a degenerated SHV and will benefit from balloon fracture before and if needed, even after implantation (Figure 6).


**Risk of coronary artery obstruction (CAO)**


During a VIV procedure, the risk of CAO is four times greater than in TAVR due to the displacement of native valve leaflets towards the coronary ostia. The factors to consider are SHV type, coronary height, sinus of Valsalva (SOV) dimensions and VTC defined by the distance between the virtual ring of the fully expanded THV, simulated into the previous SHV and the coronary ostium. Considering these parameters, the VIVID registry investigators, as aforementioned, described three class of different risk profile (from low to high risk with VTC < 4mm and small SOV) [11]

When the estimated risk of coronary occlusion is high, the following procedural strategy may be considered:-**Lower implantation of the THV:** Preferring a **SEF** due to the possibility of re-capturing or checking coronary flow before definite deployment;-**Chimney snorkel stenting technique**: Wiring the coronary artery and putting a stent on standby to be eventually implanted after THV deployment if the coronary flow is inadequate at angiographic control (Figure 7);-**Orthotopic Snorkel Stenting Technique**: Re-cannulation and wiring after THV release to have a more physiologic stenting through the prosthesis valve frame structure [42]-**BASILICA:** Intentional laceration of surgical leaflets with an electrified guidewire to create a communication between the sinus and neo-sinus [43]. Abdel-Wahab et al. [44], in the their multicentre EURO-BASILICA registry, showed a 99% of technical success and 88.3% of procedural success with an encouraging rate of freedom from any target leaflet-related CAO (90.6%) with a low rate of total coronary obstruction (2.4%). However, it is still considered a complex interventional procedure requiring meticulous preprocedural planning, dedicated material (sometimes the use of cerebral embolic protection, CEP) and high operator expertise.

## 7. Conclusions

Fifteen years after the first case, ViV-TAVR has now become a safe and effective procedure for many patients with both failed stenotic or regurgitant bioprosthetic valves. Pre-procedural multimodality imaging assessment is crucial for the best THV selection and implantation strategy. However, some pitfalls remain to be overcome, such as device malpositioning, ostial coronary obstruction, and vascular complications. In this sense, the development of new advanced and safer technologies and growing expertise of interventionalists could be the key to optimize percutaneous treatment and make it preferential to a re-do surgery.

## Figures and Tables

**Figure 1 jcm-13-00341-f001:**
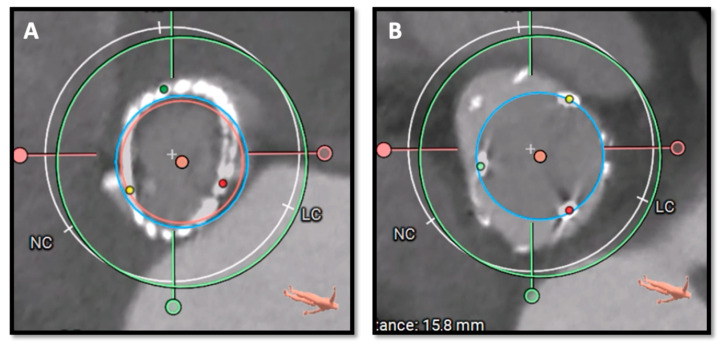
Pre-procedural CT scan simulating a 26 Sapien 3 Ultra implantation showing good sealing of the THV prosthesis to the hinge points of the Perceval bioprosthesis at different heights (LVOT in panel (**A**) and SOV in panel (**B**)). CT: computed tomography; THV, transcatheter heart valve; LVOT, Left ventricular outflow tract; SOV, Sinus of Valsalva.

**Figure 2 jcm-13-00341-f002:**
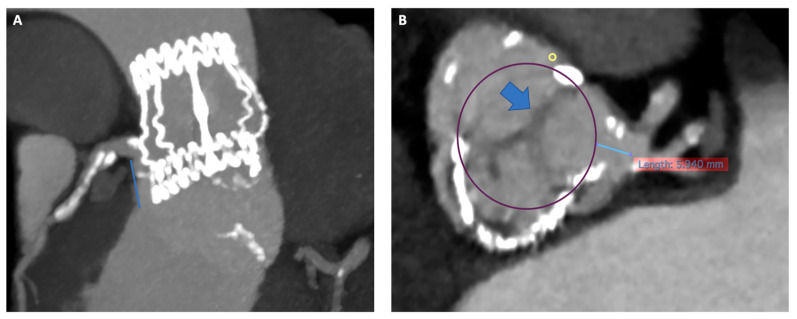
Risk of coronary artery obstruction evaluation at pre-procedural CT scan. Panel (**A**) Permissive height of LCA (18 mm). Panel (**B**) VTC of 5.9 mm simulating a 26 Sapien 3 Ultra with wide Sinus of Valsalva and redundant leaflet. CT, computed tomography; LCA; left coronary artery; VTC, valve to coronary ostia.

**Figure 3 jcm-13-00341-f003:**
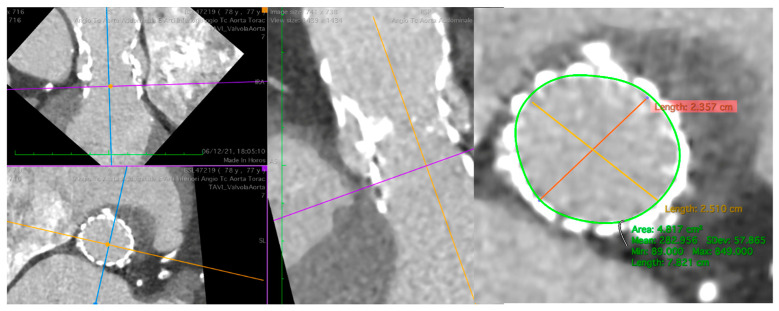
Pre-procedural CT of the aorta with a degenerated Evolut R 29. At SOV, evidence of an area of 4.8 cm^2^ and a perimeter of 78 mm suitable for a Sapien 3 23. CT, computer tomography; SOV, Sinus of Valsalva.

**Figure 4 jcm-13-00341-f004:**
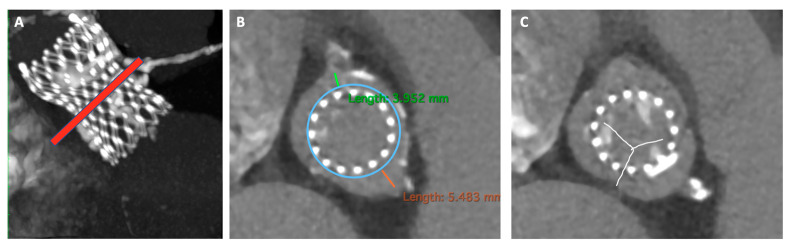
Risk assessment of coronary obstruction in redo-TAVR. Panel (**A**) Determination of the risk plane which corresponds to the top of the neo-skirt (red line). Panel (**B**) Estimated VTC distance, respectively of 3.9 mm and 5.4 mm. Panel (**C**) Leaflets overhang aiming at node 6 implantation (as low as possible). VTC, valve to coronary ostia.

**Figure 5 jcm-13-00341-f005:**
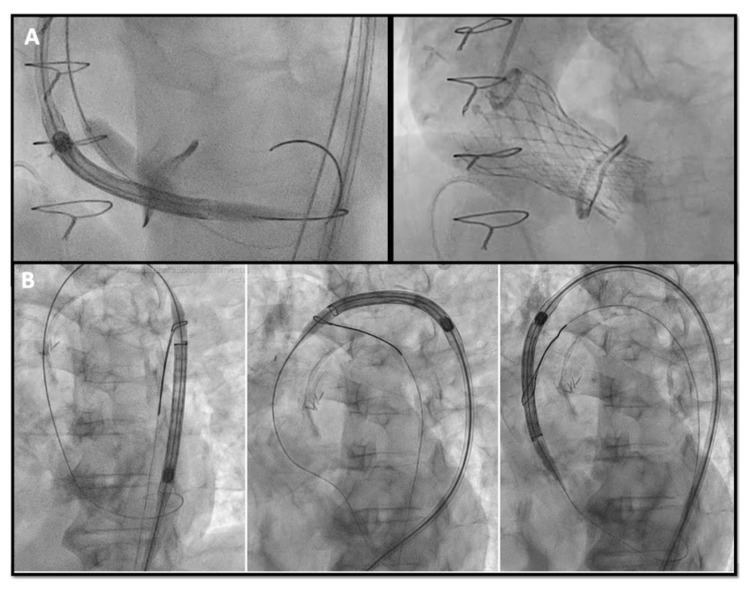
(**A**) buddy-balloon technique with VACS II 16 mm balloon inflation parallel to the Corevalve delivery to perform ViV in a Sorin Mitroflow 23 mm; (**B**) transfemoral TAVR with snare technique in a severe calcific native aortic valve stenosis VIV: valve-in-valve; TAVR: transcatheter aortic valve replacement.

**Figure 6 jcm-13-00341-f006:**
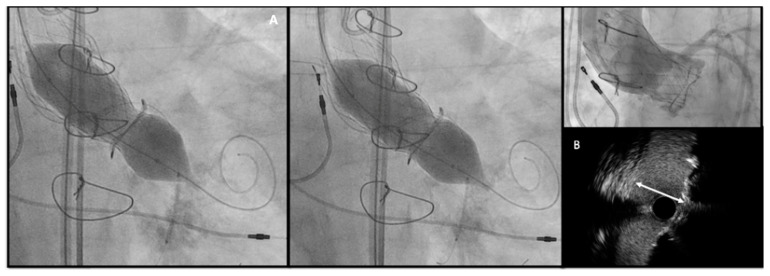
(**A**) Bioprosthetic valve fracture with a True Balloon 20 balloon after valve-in-valve with Corevalve Evolut Pro + n.23 implantation in a Mitroflow 19 (LCA protection without final stent implantation) (**B**) IVUS shows patency of the VS with a minimum distance between aortic wall and prosthetic leaflet of 2.2 mm at STJ. LCA: left coronary artery; IVUS: intra-vascular ultrasound; STJ: sino-tubular junction distance.

**Figure 7 jcm-13-00341-f007:**
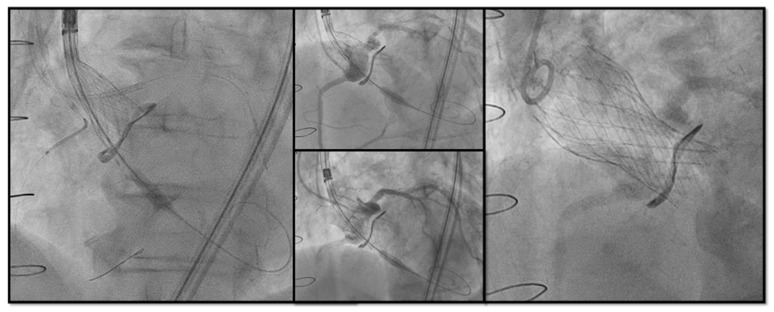
Double Chimney Stenting Technique with 4.0 × 33 Everolimus eluting stent implantation at ostial RCA and 4.0 × 18 at proximal left main before Corevalve Evolut Pro + 23 mm valve-in-valve implantation in a Mitroflow 23.

**Table 1 jcm-13-00341-t001:** Challenges of TAVR in SAVR.

Type of SAVR	Characteristic	Challenges in ViV
Stented - Internally mounted leaflets - Externally mounted leaflets Stentless	Smaller ID Wider ID Wider ID	High residual gradients Patient prosthesis mismatch Coronary obstruction Lack of fluoroscopic markers

**Table 2 jcm-13-00341-t002:** Factors related to Index THV to consider when planning redo-TAVR.

Valve Characteristics	
Type of Valve	BEV, MEV, SEV
Stent frame height	Low/high
Leaftet position	Intra-annular/supra-annular
Skirt length	
Stent Expansion	Hypo-expanded/norm expanded
Leaflets deflection	
**Anatomical characteristics**	
Implant height	Low/high
Commissural alignment to ostia	
Distance coronary ostia-stent frame	
Size of native root	
Valve position in relation to annular plane	Straight/canted

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
