# Peer review of "Valve-in-Valve Transcatheter Aortic Valve Replacement: From Pre-Procedural Planning to Procedural Scenarios and Possible Complications"

_jcm, 2024, doi:10.3390/jcm13020341_

Round 1
Reviewer 1 Report
Comments and Suggestions for Authors
This is a narrative review of Valve-in-valve transcatheter aortic valve replacement.
The review is not well structured methodologically, does not follow a coherence and the aspects described do not present criticism which affects the relevance of a topic, which has been much explored lately. The text is not formatted according to the requirements of the newspaper and presents several grammatical errors and fluency of the English language, I believe that a review by a native of the language would be very important.
The text presents weaknesses in its content and does not bring relevance to the reader, one could improve the content with a deeper review or starting from an essential question within the topic and develop the narrative with more up-to-date and relevant aspects.
Comments on the Quality of English Language
The text is not formatted according to the requirements of the newspaper and presents several grammatical errors and fluency of the English language, I believe that a review by a native of the language would be very important.
Author Response
We are thankful to the Editor and to the Reviewers for the thorough review and for the opportunity to submit a revised version of our manuscript entitled “Valve-in-valve transcatheter aortic valve replacement: from pre-procedural planning to procedural scenarios and possible complications " (jcm-2762846).
In this revised version of the manuscript, we have addressed the insightful comments of the editorial committee and of the external reviewers. We believe that changes requested and made to the manuscript have significantly improved the quality of our paper.
All changes to the manuscript are highlighted in red within the text.
In this reply letter, each comment by the reviewers (bold) is followed by our response. For substantive changes made to the manuscript, we provide a description of what we did and where. Important sentences, paragraphs or sections in response to the comments have been included in italics.
Reviewer #1
We thank the reviewer for the consideration regarding the manuscript.
Review has been revisited enhancing the coherence of the exposed concepts and delving more deeply into the latter, paying particular attention to the strategies available today for the management of these patients and carefully discussing possible decision-making hubs.
In agreement with Reviewer’s observations and notes we revised the language with the help of a native English speaker
Reviewer 2 Report
Comments and Suggestions for Authors
Di Muro et al. report on the role of valve-in-valve transcatheter aortic valve replacement for failed surgical and transcatheter bioprosthetic heart valves and describe the current clinical evidence, pre-procedural planning, procedural techniques, and potential complications. As the indications for TAVR expand to include younger patients and those at low or moderate surgical risk, the long-term durability of transcatheter heart valves is becoming increasingly important.This is a very meaningful work as it provides better guidance for clinical work.
Here are some comments.
1. The language and format need further audits, e.g. P3 line 64.
2. Can you add a flowchart or graphical abstract to describe the entire process of valve-in-valve transcatheter aortic valve replacement from preoperative preparation to postoperative complications?
3. Label Table 1 and Table 2 accordingly in the text.
4. Can you elaborate on your conclusions?
Comments on the Quality of English Language
1. The language and format need further audits, e.g. P3 line 64.
Author Response
We are thankful to the Editor and to the Reviewers for the thorough review and for the opportunity to submit a revised version of our manuscript entitled “Valve-in-valve transcatheter aortic valve replacement: from pre-procedural planning to procedural scenarios and possible complications " (jcm-2762846).
In this revised version of the manuscript, we have addressed the insightful comments of the editorial committee and of the external reviewers. We believe that changes requested and made to the manuscript have significantly improved the quality of our paper.
All changes to the manuscript are highlighted in red within the text.
In this reply letter, each comment by the reviewers (bold) is followed by our response. For substantive changes made to the manuscript, we provide a description of what we did and where. Important sentences, paragraphs or sections in response to the comments have been included in italics.
Reviewer #2
We thank the reviewer for the consideration regarding the manuscript.
In agreement with his observations and notes we revised the language with the help of a native English speaker (1).
You will find a graphical abstract describing phase by phase the entire process of valve-in-valve transcatheter aortic valve replacement from pre-procedural planning to possible complications as you gently suggested. We hope you will appreciate it (2)
We also labelled respectively Table 1 and 2 in the manuscript(3).
Conclusions have been developed following your suggestion (4).
Reviewer 3 Report
Comments and Suggestions for Authors The manuscript is interesting and centered on a hot topic in literature.
The main topic of this review is undoubtedly appealing and interesting for
the scientific community. The structure of the article is clear and reader-friendly. Some points need to be modified. First of all a focus on current guidelines concerning the decision-making process for young patients, for
which TAVR is not first line. In the manuscript you this issue is not discussed in an exhaustive way.
It would be interesting to analyze the reasons beyond the choice of a bioprosthetic valve even for patients not suitable (younger or less comorbidity burdened), maybe comparing the long-term outcomes
of patients implanted with a mechanical valve versus patients who underwent ViV-TAVR, with respective benefits and risks.
Eventually, have to be added a paragraph concerning the state of the art, according to European and American guidelines, for redo SAVR and ViV-TAVR, looking even at the main suggestions from the most relevant consensus statements.
Concerning the structure of the mauscript:
Please add at least 3 keywords.
Line 68: too strong sentence, no trial data available, too short term data showed. Avoid personal sentences or use them starting with “in our opinion”, especially if these sentences are not supported by clear and indisputable results
Line 72 and 74: redoSAVR is still the first line treatment, not “might be considered”
Line 128: not clear the meaning and the sequence of commas and brackets
Line 163: no data to allow this sentence, remove it
Line 248: please reduce the dimension of the text
Line 253: please add a reference after the Vivid registry
Line 256: it is part of the list? In that case add a high dash
Line 270: (39, 40)
Figure 5 and 7: please put a number inside the figure, so you can use them and not “second panel”… it would be more readable Comments on the Quality of English Language The article is well-written, some mistakes must be correct:
Line 54 and all over the manuscript: no comma before and
Line 141: bioprosthesis
Line 145: malpositioning
Line 200: the risks raise
Line 273: will benefit
Line 312: the sentence is not clear, the subject vary; use a better English structure
Author Response
We are thankful to the Editor and to the Reviewers for the thorough review and for the opportunity to submit a revised version of our manuscript entitled “Valve-in-valve transcatheter aortic valve replacement: from pre-procedural planning to procedural scenarios and possible complications " (jcm-2762846).
In this revised version of the manuscript, we have addressed the insightful comments of the editorial committee and of the external reviewers. We believe that changes requested and made to the manuscript have significantly improved the quality of our paper.
All changes to the manuscript are highlighted in red within the text.
In this reply letter, each comment by the reviewers (bold) is followed by our response. For substantive changes made to the manuscript, we provide a description of what we did and where. Important sentences, paragraphs or sections in response to the comments have been included in italics.
Reviewer #3
We thank the reviewer for the comments regarding the manuscript.
In agreement with his observations and notes we revised the language with the help of a native English speaker.
We welcomed the invitation to develop the introduction section by expanding the part on the indications for choosing between mechanical and biological aortic prosthesis and discussing strength, weakness and perspectives of these strategies; we hope you will now appreciate.
We have also reorganized the paragraph on the indications to re-do SAVR vs redo TAVR so that they are clearer and immediate to the readers.
We are thankful for the accurate comments regarding the structure of the manuscript and the quality of English language; comments were addressed in the revised manuscript.
Personal considerations were deleted.
You will find all these corrections in red within the text.
Round 2
Reviewer 1 Report
Comments and Suggestions for Authors
The article has improved significantly, however it would be necessary to revise the conclusion, it is together with future perspectives that should be in the discussion, and the citation of figures is not correct in the conclusion. The conclusion should be concise and answer your research question.
Author Response
I express my gratitude to the reviewer for his thoughtful comments.
We have enhanced the clarity of our conclusions by removing 'future perspectives' and endeavored to respond to the review question in a more concise and direct manner.
Additionally, we have reorganized the figure numbering, placing those associated with pre-procedural planning in the text before those related to the section on procedural management.
All modifications are highlighted in red within the text and we trust that you will find them satisfactory.
Reviewer 3 Report
Comments and Suggestions for Authors
All the previous suggestions were followed.
Now the article is complete and not misleading.
Author Response
We thank the reviewer for his final judgment